# Predicting 90-Day Prognosis in Ischemic Stroke Patients Post Thrombolysis Using Machine Learning

**DOI:** 10.3390/jpm13111555

**Published:** 2023-10-30

**Authors:** Ahmad A. Abujaber, Ibrahem Albalkhi, Yahia Imam, Abdulqadir J. Nashwan, Said Yaseen, Naveed Akhtar, Ibraheem M. Alkhawaldeh

**Affiliations:** 1Nursing Department, Hamad Medical Corporation, Doha P.O. Box 3050, Qatar; 2College of Medicine, Alfaisal University, Riyadh 11533, Saudi Arabia; 3Department of Neuroradiology, Great Ormond Street Hospital NHS Foundation Trust, Great Ormond St., London WC1N 3JH, UK; 4Neurology Section, Neuroscience Institute, Hamad Medical Corporation, Doha P.O. Box 3050, Qatar; 5School of Medicine, Jordan University of Science and Technology, Irbid 22110, Jordan; 6Faculty of Medicine, Mutah University, Al-Karak 61710, Jordan

**Keywords:** ischemic stroke, thrombolysis, machine learning, prognosis prediction, 90-day modified ranking scale

## Abstract

(1) Objective: This study aimed to construct a machine learning model for predicting the prognosis of ischemic stroke patients who underwent thrombolysis, assessed through the modified Rankin Scale (mRS) score 90 days after discharge. (2) Methods: Data were sourced from Qatar’s stroke registry covering January 2014 to June 2022. A total of 723 patients with ischemic stroke who had received thrombolysis were included. Clinical variables were examined, encompassing demographics, stroke severity indices, comorbidities, laboratory results, admission vital signs, and hospital-acquired complications. The predictive capabilities of five distinct machine learning models were rigorously evaluated using a comprehensive set of metrics. The SHAP analysis was deployed to uncover the most influential predictors. (3) Results: The Support Vector Machine (SVM) model emerged as the standout performer, achieving an area under the curve (AUC) of 0.72. Key determinants of patient outcomes included stroke severity at admission; admission systolic and diastolic blood pressure; baseline comorbidities, notably hypertension (HTN) and coronary artery disease (CAD); stroke subtype, particularly strokes of undetermined origin (SUO); and hospital-acquired urinary tract infections (UTIs). (4) Conclusions: Machine learning can improve early prognosis prediction in ischemic stroke, especially after thrombolysis. The SVM model is a promising tool for empowering clinicians to create individualized treatment plans. Despite limitations, this study contributes to our knowledge and encourages future research to integrate more comprehensive data. Ultimately, it offers a pathway to improve personalized stroke care and enhance the quality of life for stroke survivors.

## 1. Introduction

Stroke poses a significant worldwide health challenge, ranking as the second-leading cause of mortality and the third-leading contributor to disability and mortality combined [1]. On a global scale, the occurrence and prevalence of strokes were more pronounced in females than in males, even though there were no variations in the number of stroke-related fatalities between the two sexes [2].

Research has consistently shown that ischemic stroke occurs more frequently than hemorrhagic stroke (ICH) [3]. Over time, various reperfusion methods, including mechanical and intravenous approaches, have demonstrated improved results in treating individuals with acute ischemic stroke (IS) [3]. Intravenous thrombolysis using tissue plasminogen activator (t-PA) is considered a standard IS treatment, proving both effective and safe when administered within 3–4.5 h from the onset of stroke [4]. Additionally, IS patients who have received thrombolytic therapy have shown more favorable outcomes in terms of dementia and mortality compared to those who have not [5,6].

Qatar stands out for its diverse population, with over 2.7 million residents, including approximately 300,000 native Qataris. Interestingly, there is a higher proportion of females among the Qatari population, and the demographic makeup of Qatar is characterized by a significant presence of young, non-disabled individuals hailing from South/Southeast Asia, representing various ethnic backgrounds [7]. Imam and colleagues reported that among the local Qatari population, the average stroke incidence is 92.04 per 100,000 adults, with stroke typically occurring around the age of 64. Notably, local Qatari women have a higher susceptibility to stroke and disability compared to their male counterparts [7].

Hamad General Hospital (HGH) serves as Qatar’s principal tertiary healthcare facility and is the country’s central hub for acute stroke services. From 2014 to mid-2022, HGH’s stroke services successfully administered 971 thrombolysis procedures for individuals with ischemic stroke (IS); all were treated with t-PA. Aligned with the global trend, this accounts for approximately 6% of the total stroke patient population treated at the hospital. Although roughly 25% of stroke patients may be eligible for thrombolytic treatment worldwide, 2–10% of all stroke patients typically undergo thrombolysis [8]. HGH’s stroke service maintains an active prospective registry that records crucial clinical and demographic information about stroke patients, including their complications and outcomes [9].

Predicting stroke outcomes is vital in clinical care, and one widely used metric for assessing post-stroke disability and independence is the 90-day modified Rankin Scale (mRS) score after hospital discharge. This scale serves as a crucial tool for evaluating the expected post-stroke outcomes [10]. Despite the substantial body of literature addressing short-, intermediate, and long-term stroke outcome predictions, there has been notably limited research into forecasting the outcomes and functional prognosis after thrombolytic treatment.

Many predictive scores and scales were used to predict prognosis and to forecast the 90-day mRS, such as the National Institutes of Health Stroke Scale (NIHSS) scores and other composite scores such as the Five Simple Variables (FSV) score, the PLAN (Pre-admission dependence and comorbidities, Level of consciousness, Age, and focal Neurological deficit), or a composite of clinical prediction scores. These predictions are influenced by several factors, including patient demographics, clinical traits, stroke severity (incorporating the NIHSS), treatment approaches, age, sex, previous stroke history, and smoking habits [11,12,13].

Similarly, several scoring systems were consolidated to create a more practical and user-friendly assessment tool known as the modified SOAR (mSOAR) score, taking into account stroke subtype, Oxfordshire Community Stroke Project classification, age, pre-stroke modified Rankin score (mRS), and the National Institutes of Health Stroke Scale score. This score has demonstrated its effectiveness in forecasting post-stroke disability [14]. Clinical indicators like door-to-needle time (DNT) and infarct size were found to be valuable predictors of treatment outcomes. A shorter DNT and a smaller infarct size are linked to more positive outcomes [15,16].

Insights from the ECASS III trial, which evaluated alteplase treatment in the 3–4.5-h window, have revealed important stroke prognostication; it demonstrated that the alteplase group had better outcomes (measured by a 90-day mRS score) compared to the placebo group [3]. Additionally, patients who received t-PA showed improved three-year survival rates compared to those who did not [6]. However, thrombolytic therapy was associated with increased early deaths (within 7–10 days of treatment) due to post-t-PA intracranial hemorrhage. It is worth noting that thrombolytic treatment yields more favorable outcomes when administered within a 3 h timeframe from the stroke onset compared to those receiving treatment beyond the 3 h window [17].

Another study revealed that patients who received intravenous thrombolysis treatment outside the recommended optimal window of 3–4.5 h from the onset of stroke symptoms, or in cases where the precise onset time was unknown, showed enhanced 90-day post-stroke mRS. However, this particular group also had a higher mortality rate when compared to individuals who did not undergo this treatment [18]. More recently, given that different patients have different physiology, it was noted to be feasible and safe, to have improved neurological function acutely, and to have increased the number of patients with an mRS of 0–1 [19].

According to Deb-Chatterji and colleagues, younger patients (below 49 years of age) who received thrombolysis treatment showed better prospects in terms of their prognosis when evaluated using the mRS scale, as opposed to older patients [20].

Machine learning algorithms have recently been employed to forecast stroke outcomes and prognosis, demonstrating comparable or superior effectiveness to conventional methods like logistic regression [21]. Predicting stroke outcomes is a complex task that involves numerous variables. Hence, incorporating machine learning models to augment predictive capabilities could further advance the precision medicine agenda. The remarkable computational prowess of machine learning has the potential to enhance the prognostic process, facilitating personalized treatment strategies, enhancing patient outcomes, and alleviating the global burden of stroke.

The primary motive behind this study is to harness the predictive capabilities of machine learning to uncover meaningful and actionable connections between various study variables and their impact on patient outcomes. By doing so, we empower clinicians in Qatar and similar settings with a robust tool for crafting individualized care plans tailored to the unique context of their patients. Specifically, our study aims to construct a machine learning model to predict the prognostic outcomes of thrombolytic (fibrinolytic) treatment. This model will delve into the multitude of factors influencing the prognosis as measured by the 90-day mRS after discharge for patients who undergo thrombolysis for ischemic stroke.

## 2. Methods

This study obtained approval from the Institutional Research Board (IRB) of Hamad Medical Corporation, Doha, Qatar, under reference MRC-01-22-594.

### 2.1. Data Collection

Data were sourced from the Stroke Registry at Hamad General Hospital (HGH), encompassing January 2014 to July 2022. The dataset encompassed all adults aged 18 years and above admitted to HGH with a primary stroke diagnosis, including cases of IS, TIA, ICH, and stroke mimics. During the study period, 15,859 patients sought specialized stroke care at the hospital, 971 received thrombolytic treatment, and 723 patients were ultimately included in this study.

### 2.2. Inclusion/Exclusion Criteria

From the initial 15,859 patients, 971 adults (≥18) were diagnosed with IS and underwent thrombolysis treatment. Of them, 948 were discharged alive from the hospital. All patients who missed their 90-day follow-up appointment or had unstandardized 0–6 mRS scores were excluded, resulting in 723 patients eligible for analysis. Figure 1 summarizes the inclusion/exclusion procedure.

### 2.3. Baseline Variables

The collected variables encompassed a wide range of patient characteristics, including demographic details, ethnicity, mode of arrival, time from stroke onset to hospital admission, initial vital signs (comprising systolic blood pressure (SBP), diastolic blood pressure (DBP), and heart rate (HR)), stroke risk factors, pre-existing medical conditions, door-to-needle time (DNT), hospitalization outcomes (including hospital-acquired infections like pneumonia and urinary tract infections (UTIs)), and stroke severity. Stroke severity upon admission was measured using the NIHSS [22,23]. Additionally, the mRS was collected upon admission, which assesses the patient’s pre-stroke condition based on family member reports and is rated on a scale of 0–6 [21]. The etiology of ischemic stroke was determined using the Trial of Org 10,172 in Acute Stroke Treatment (TOAST) classification [24]. Body Mass Index (BMI) categories were established using the CDC’s 5-class definition for adult overweight and obesity [25]. Information regarding patients’ prescribed medications, such as anti-platelet agents and anticoagulants, was documented in the registry and cross-referenced with electronic health records (EHRs) during hospital stays. Additionally, fundamental laboratory results, including Prothrombin Time (PT), Partial Thromboplastin Time (PTT), and Random Blood Sugar (RBS), were included in this study.

With respect to ethnicity, patients were categorized into five groups based on their stated nationality: Qatari, Middle East and North Africa (MENA) region, South Asia region, Southeast Asia region (following the United Nations geo-scheme), and all other nationalities grouped as ‘other’ [26,27,28]. It is important to note that Qatari patients were placed in a distinct category to enable meaningful comparisons. This consideration considers Qatar’s unique demographic composition, where most of the population consists of expatriates [7,29]. This approach aligns with the methodology consistently employed in previous studies examining strokes in Qatar [9,26]. All recorded risk factors, including comorbidities and smoking history, were meticulously confirmed during the patient’s hospitalization and cross-verified by the stroke registry staff through EHRs. Subsequently, 28 input variables were included in this study as summarized in Table 1.

### 2.4. Outcome Variable

The 90-day post-discharge mRS was transformed into a binary variable for simplicity. An mRS score of ≤2 was considered favorable, indicating a positive prognosis, while an mRS score > 2 was deemed unfavorable, signifying an unfavorable prognosis [7,21,30].

### 2.5. Handling Missing Data

When data values were absent, we utilized the Multiple Imputation using the Chained Equations (MICE) method, which is considered one the most up-to-date and trusted methods to handle missing data [31,32]. Among the dataset variables, it was noted that Partial Thromboplastin Time (PTT) had the highest percentage of missing data at 5.95%, followed by Prothrombin Time (PT) at 5.25%, platelets count at 2.1%, and HR, RBS, SBP, and DBP all having less than 1% missing data. The percentage of missing data is less than 6%, which, compared to many other studies in the literature (8–24%), is considered low [33,34]. The cohort exhibited an unfavorable mRS score (>2) rate of 47.23%, indicating a reasonably class-balanced dataset.

### 2.6. Model Training and Evaluation

The dataset was partitioned into a training set (80%) and a validation set (20%) using stratified random sampling. The training dataset was used to build the models, and the performance was assessed using the validation dataset. Five machine learning models were trained, including XGBoost Classifier, Random Forest (RF), Support Vector Machine (SVM), logistic regression (LR), and Classification and Regression Tree (CART).

Various classification metrics were employed to evaluate the models’ effectiveness, including accuracy, precision, specificity, recall, F1-score, area under the receiver operating characteristic curve (AUC), Matthew’s correlation coefficient (MCC), log loss, and Brier score [35,36,37,38,39]. These metrics offer insights into the model’s ability to accurately classify both positive cases (patients with favorable outcomes) and negative cases (patients with unfavorable outcomes). Model performance results are detailed in Table 2. The model achieving the highest F1 score will be selected as the primary model for subsequent external and temporal validation.

SHAP (SHapley Additive exPlanations) is a robust library employed to interpret the predictions made by machine learning models [40]. This tool generates feature importance scores at the individual level, known as SHAP values. These values quantify the impact of each feature on a specific prediction result. We used SHAP analysis to identify the key predictors.

## 3. Results

### 3.1. Model Evaluation

As presented in Table 2 and Figure 2, the models delivered satisfactory performances, with the SVM model emerging as the top performer. SVM achieved an AUC of 0.76, an accuracy of 0.70, a precision of 0.6, a recall of 0.68, and an F1-score of 0.64. The XGBoost model also demonstrated competitive results, boasting an F1-score of 0.64 and AUC of 0.756, although it had a slightly lower MCC (0.36) and a comparable recall (0.72). In contrast, logistic regression lagged with an accuracy of 0.66 and AUC of 0.72. The Decision Tree Classifier performed least effectively, achieving an accuracy of 0.62 and AUC of 0.62 with lower precision (0.51) and specificity (0.61), resulting in a lower overall score of 0.57.

These results underscore the superiority of the XGBoost and SVM models, which exhibited robust performance across multiple evaluation metrics. In contrast, the Decision Tree model fell short due to its limited accuracy and precision. As a result, the SVM is selected as the primary model for subsequent SHAP analysis, external validation, and temporal validation in predicting functional outcomes.

### 3.2. SHAP Analysis

The SHAP analysis yielded invaluable and pivotal insights for forecasting functional outcomes (mRS90). In this context, the crucial factors, ranked by importance, are stroke severity upon presentation (measured by NIHSS), pre-stroke mRS, RBS, SBP, DBP, HTN, CAD, SUO, and UTI. Higher values of NIHSS at admission and pre-stroke mRS scores were positively correlated with unfavorable outcomes, underscoring the impact of initial neurological impairment. Additionally, elevated blood pressure (systolic and diastolic) and blood sugar levels were associated with a less favorable prognosis. The subtype of ischemic stroke, pre-existing comorbid conditions, and the development of hospital-acquired UTI also significantly correlated with the prognosis of patients who underwent thrombolysis.

## 4. Discussion

This study delved into the effectiveness of five machine learning models and employed SHAP analysis to pinpoint the key predictors influencing the 90-day mRS in ischemic stroke patients who underwent thrombolysis procedures. The aim was to shed light on the accuracy of machine learning models in predicting prognosis in thrombolyzed ischemic stroke patients. The limited existing literature discussing this important aspect of stroke research positions our study as one of the few in this field, highlighting its contribution and originality.

Among the different models chosen here, the results underscore the superiority of the SVM model when compared to other models. The SHAP analysis on the SVM model revealed critical variables that enhance its predictive capabilities. These influential variables encompass stroke severity (measured by NIHSS at admission), pre-stroke mRS, admission RBS, blood pressure (both systolic and diastolic) at admission, known comorbidities, notably HTN and CAD stroke subtype as per TOAST classification, and hospital-acquired UTI. It is worth noting that the remaining variables demonstrated low SHAP values, indicating their limited contribution to the model’s predictive performance. Importantly, the identified key predictors, apart from hospital-acquired UTI, can serve as early indicators of an unfavorable prognosis, as they can be assessed upon admission. As a result, this research has the potential to enhance clinicians’ ability to predict the prognosis of IS patients undergoing prompt thrombolysis and tailor care strategies to address prognosis risk [41]. Furthermore, it may assist families in forming realistic expectations regarding expected outcomes, thereby improving their engagement in the care delivery plan [42].

Consistent with prior research, stroke severity at the time of presentation, as assessed by NIHSS, emerged as the most significant predictor of mRS at 90 days [43,44]. SHAP analysis corroborated that NIHSS at admission exerted the most substantial influence on the model’s predictive accuracy. In this study, the average NIHSS score was 10.3 ± 5.9. Notably, it was observed that the mean NIHSS score for patients experiencing an unfavorable prognosis was significantly higher than that of the favorable group (12.6 vs. 8.2), *p*-value < 0.05.

The pre-stroke modified Rankin Score (mRS) plays a pivotal role in predicting the prognosis of individuals with IS who underwent thrombolysis. This finding aligns with Goda et al.’s findings, indicating that patients who reported an unfavorable 90-day mRS had a notably higher average pre-stroke mRS score: 0.32 vs. 0.08, *p*-value < 0.05 [45]. This underscores the significance of assessing the functional status of patients prior to the onset of a stroke. Such assessments enable healthcare providers to plan for preventive measures tailored to individuals at risk of a poor prognosis during their treatment journey.

The third factor that enhances the model’s predictive accuracy is the RBS level obtained at the presentation. This study corroborates previous research by demonstrating a correlation between higher RBS levels at admission and an unfavorable prognosis [46]. In this study, the mean RBS was 9.3 ± 4.6 mmol/L, and the average RBS of patients with an unfavorable mRS at 90 days was significantly higher than that of the favorable group (9.9 vs. 8.7, with a *p*-value below 0.05).

Likewise, BP recorded upon admission to the emergency department has emerged as a predictive factor for stroke mortality [47,48,49]. SHAP analysis revealed that both SBP and DBP play essential roles in the model’s predictability. However, the SHAP value of SBP is higher than that of DBP (as illustrated in Figure 3a,b). The mean SBP and DBP were 152.4 ± 27.7 mmHg and 90.11 ± 16.9 mmHg, respectively. Elevated levels of both variables were associated with an unfavorable prognosis. Patients who presented with an unfavorable mRS 90 days after discharge had a significantly higher mean SBP at admission to the emergency room following their stroke onset compared to those with a favorable mRS90 (154.8 vs. 150.2 mmHg), *p*-value < 0.05. Similarly, DBP exhibited a similar relationship with the mRS90 (91.5 vs. 88.8 mmHg, respectively, *p*-value < 0.05). This is in agreement with prior reports that showed that SBP influences prognosis in a U-shaped curve in ischemic stroke in general, and the early attainment of SBP levels 140–110 post thrombolysis positively correlated with good outcomes [50,51].

Similarly, patients with pre-existing comorbidities, particularly HTN and CAD, who underwent thrombolysis exhibited a higher inclination toward an unfavorable mRS90. The SHAP analysis (as depicted in Figure 3a,b) illustrates that patients with known HTN and CAD are more likely to experience an unfavorable mRS90 [21,52,53]. It is worth noting that HTN has a higher SHAP value than CAD, indicating a more significant impact on the model’s performance, as demonstrated in Figure 3a,b. It is observed that over 50% of patients with HTN reported an unfavorable mRS90, in contrast to 40.6% of patients without a history of HTN, *p*-value < 0.05. Similarly, 57.1% of patients with CAD reported an unfavorable mRS90, compared to 45.6% without CAD, *p*-value < 0.05. This finding provides insight that the treating provider might consider certain precautionary measures early to prevent the poor prognosis of patients with pre-existing HTN and CAD.

The subtype of ischemic stroke (IS) was identified as a factor influencing the prognosis of IS in patients who underwent thrombolysis. Recognizing the specific stroke subtype is crucial for determining appropriate treatment strategies and predicting potential treatment outcomes. Wei and colleagues reported that Stroke of Undetermined Origin (SUO) is associated with a higher mortality rate [54]. Importantly, the mechanism through which SUO impacts stroke prognosis remains poorly understood [55]. This study also revealed that 60.4% of patients with SUO reported an unfavorable mRS90, followed by 57.6% of patients with Large Vessel Disease (LVD), in contrast to 31.2%, 41.2%, and 42.7% for the stroke of determined origin (SDO), Small Vessel Disease (SVD), and embolic stroke subtypes, respectively, with a *p*-value < 0.05.

Lastly, the occurrence of hospital-acquired UTI emerges as a robust predictor of an adverse stroke prognosis. Stroke patients are recognized to be highly vulnerable to infections during their hospital stay, which can negatively impact their functional recovery [56]. Our study revealed that over 90% of patients who experienced hospital-acquired UTI reported an unfavorable mRS90, in contrast to 45.1% of patients who did not develop hospital-acquired UTI, *p*-value < 0.5.

It is crucial to emphasize that, except for hospital-acquired UTI, all the discussed predictors function as early indicators (at admission) for forecasting the prognosis of ischemic stroke in patients who have undergone thrombolysis, enhancing this study’s significance. Early prediction of the disease and treatment outcomes holds profound importance as it equips clinicians with the tools to craft individualized care strategies addressing the pivotal factors contributing to adverse outcomes, ultimately elevating the quality of life for stroke patients. Notably, our research underscores the superior performance of the SVM model compared to other models tested, achieving a commendable discrimination power with an AUC of 0.72 [57]. This encouraging outcome suggests that the SVM model can be effectively integrated into clinical settings for early 90-day prognosis prediction, empowering healthcare providers to devise personalized plans that bolster preventive measures and enhance the overall well-being of stroke patients.

## 5. Limitations

There are several limitations to consider when interpreting the findings of this study. Firstly, the data used to build the model originated exclusively from a single-center stroke registry. This limitation may restrict the generalizability of the results to more diverse healthcare settings and broader populations, although the population studied was diverse, with multiple ethnicities included. Also, the absence of certain variables, such as imaging data, may have excluded crucial predictors that influence stroke prognosis prediction.

The retrospective nature of this study introduces inherent biases and challenges in capturing real-time clinical nuances. Relying on historical data may not fully account for evolving treatment approaches and changing patient demographics, potentially affecting the model’s adaptability to clinical scenarios. The 90-day horizon for prognosis may not encompass longer-term outcomes, limiting its applicability in assessing extended prognostic trends.

The presence of missing data in some important predictors presents challenges during model development and evaluation. Although efforts were made to address this issue through the MICE technique, there remains a potential risk of bias in model performance.

While the model’s performance demonstrates promise, with moderate accuracy, precision, and recall rates, there is room for further refinement. Utilizing machine learning models introduces inherent complexity, possibly complicating the clinical interpretation and integration into practical healthcare workflows. Additionally, the model’s performance could be influenced by the absence of key features not routinely captured in the registry, suggesting the need to improve data inclusion in future research.

Lastly, while SHAP analysis identifies key predictors, this study does not establish definitive causal relationships between these variables and stroke mortality. Consequently, while this research enhances our understanding of stroke prognosis prediction, it is crucial to interpret its implications within the context of these limitations.

## 6. Conclusions

This study delves into ischemic stroke prognosis prediction in patients who have undergone thrombolysis, shedding light on critical factors influencing their 90-day outcomes. While acknowledging its limitations, this study’s findings hold significant implications for clinical practice and patient care.

Our investigation underscores the importance of early prediction as a cornerstone of effective stroke management. We have identified a set of predictors, including stroke severity at admission, pre-stroke functional status, blood sugar levels, and blood pressure, all of which offer valuable insights into patients’ prognoses. These factors can empower clinicians to craft personalized care plans addressing the elements contributing to adverse outcomes, ultimately enhancing the quality of life for stroke survivors.

Furthermore, this study highlights the superior performance of the SVM model in prognosis prediction, exhibiting commendable discrimination power. The model’s efficacy opens the door to its integration into clinical settings, where it can serve as a valuable tool for early prediction of several stroke outcomes. This, in turn, enables healthcare providers to proactively devise tailored strategies that bolster preventive measures and improve the overall well-being of stroke patients.

In the context of this study’s limitations, this research marks a significant step forward in the quest to enhance stroke prognosis prediction. It encourages future research to expand and improve stroke registries, capturing a more comprehensive dataset that includes key variables often missing in routine registries. Such efforts will further refine our predictive models and strengthen their clinical relevance.

Finally, while this study is not without its constraints, it contributes valuable insights into ischemic stroke prognosis prediction. It underscores the importance of early assessment, leveraging the power of machine learning models like SVM and tailoring care plans to individual patient needs. Ultimately, it paves the way for more effective stroke management, promising a brighter future for stroke survivors and their families.

## Figures and Tables

**Figure 1 jpm-13-01555-f001:**
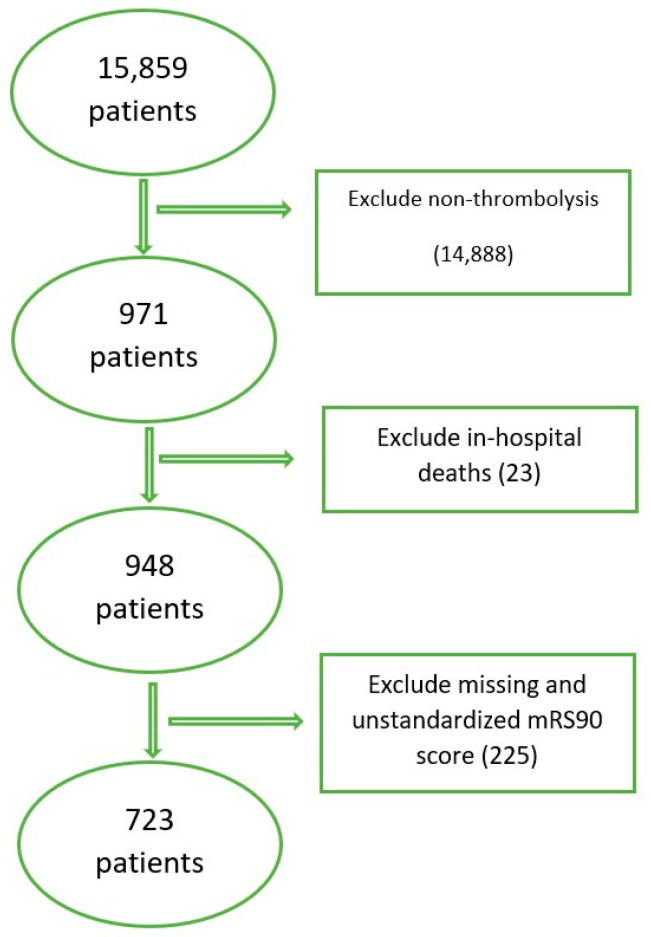
Patients’ inclusion/exclusion procedure.

**Figure 2 jpm-13-01555-f002:**
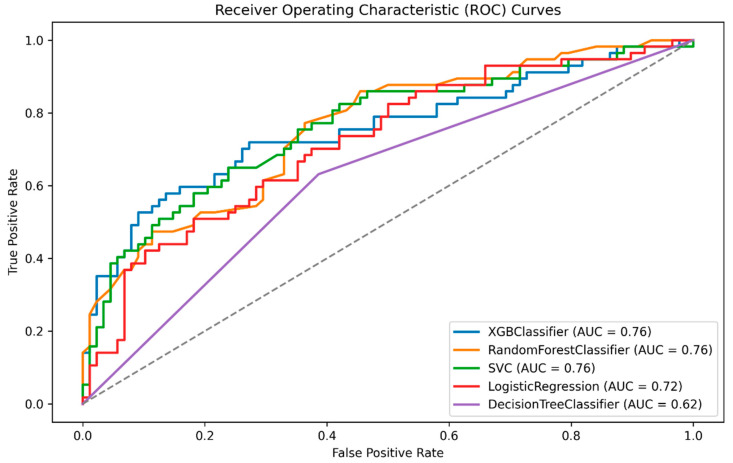
Area under the curve for the trained models.

**Figure 3 jpm-13-01555-f003:**
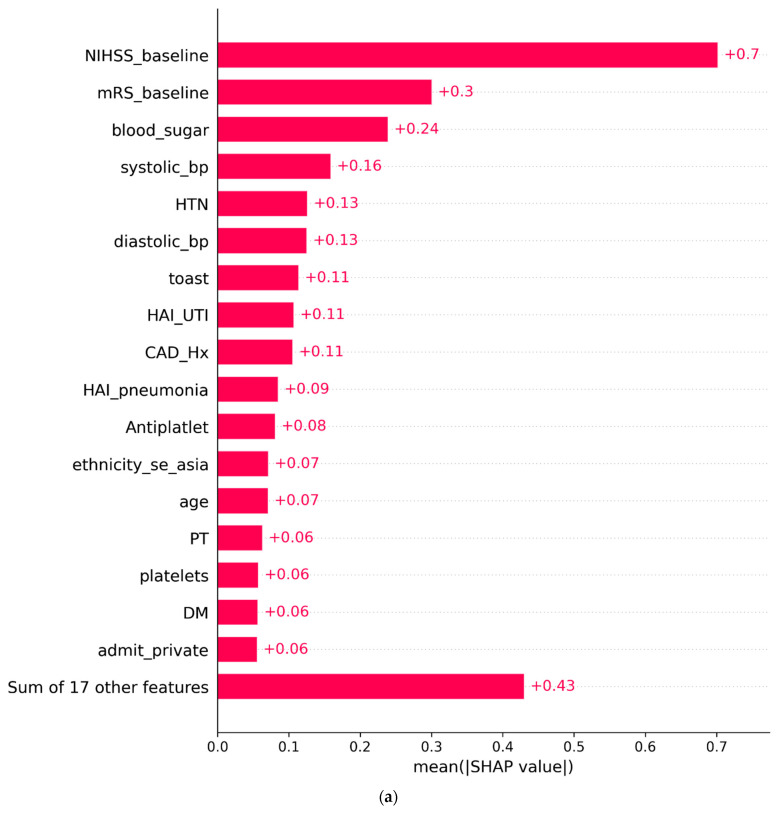
(**a**) SHAP analysis (mean SHAP value). (**b**) SHAP analysis (predictors’ impact on model output).

**Table 1 jpm-13-01555-t001:** Statistical characteristics of the collected stroke dataset.

Variable	Feature	Favorable (mRS ≤ 2)	Unfavorable (mRS > 2)	Total
Age (year)	<mean (54.1)	218	164	382
	≥mean (54.1)	163	178	341
	Mean ± SD (54.1 ± 13.1)-IQR 17			
Sex	1: Male	320	284	604
	2: Female	61	58	119
Ethnicity	1: Qatari	41	49	90
	2: MENA	77	74	151
	3: South Asian	202	178	380
	4: Southeast Asian	38	30	68
	5: Other	23	11	34
Modified Rankin Score (mRS) pre-stroke onset	<mean (0.2)	364	300	664
	≥mean (0.2)	17	42	59
	Mean ± SD (0.2 ± 0.72)-IQR 9			
NIHSS at admission	<mean (10.3)	281	147	428
	≥mean (10.3)	100	195	295
	Mean ± SD (10.3 ± 5.9)-IQR 0			
Mode of arrival	1: Ambulance	332	306	638
	2: Private vehicle	45	26	71
	3: In-hospital	4	10	14
Body Mass Index (BMI)	1: Underweight	11	10	21
	2: Normal weight	112	107	219
	3: Overweight	153	151	304
	4: Obese	82	39	121
	5: Extremely Obese	23	35	58
Time from onset to hospital arrival (hour)	1: ≤hours	347	309	656
	2: >3 h	34	33	67
Diabetes Mellitus (DM)	0: No	299	289	588
	1: Yes	82	53	135
Hypertension (HTN)	0: No	142	101	243
	1: Yes	239	241	480
Dyslipidemia	0: No	201	189	390
	1: Yes	180	153	333
Prior stroke	0: No	357	315	672
	1: Yes	24	27	51
Atrial Fibrillation (AF)	0: No	366	313	679
	1: Yes	15	29	44
Coronary artery disease (CAD)	0: No	336	282	618
	1: Yes	45	60	105
Tobacco use	0: No	277	267	544
	1: Yes	104	75	179
RBS at admission (mmol/L)	<mean (9.3)	262	206	468
	≥mean (9.3)	117	135	252
	Mean ± SD (9.3 ± 4.6)-IQR 5			
SBP at admission (mmHg)	<mean (152.4)	212	173	385
	≥mean (152.4)	168	169	337
	Mean ± SD (152.4 ± 27.7)-IQR 37			
DBP at admission (mmHg)	<mean (90.11)	225	176	401
	≥mean (90.11)	155	165	320
	Mean ± SD (90.11 ± 16.9)-IQR 22			
HR at admission (bpm)	<mean (84.3)	208	172	380
	≥mean (84.3)	168	169	337
	Mean ± SD (84.3 ± 16.2)-IQR 21			
Anti-platelets	0: No	310	269	579
	1: Yes	71	73	144
Anticoagulants	0: No	362	323	685
	1: Yes	19	19	38
Platelet count at admission	<mean (259.5)	201	178	388
	≥mean (259.5)	171	149	320
	Mean ± SD (259.5 ± 78.15)-IQR 88			
Prothrombin Time (PT)	<mean (10.4)	192	150	342
	≥mean (10.4)	166	177	343
	Mean ± SD (10.4 ± 4.5)-IQR 2.4			
Partial Thromboplastin Time (PTT)	<mean (26.5)	218	203	421
	≥mean (26.5)	138	121	259
	Mean ± SD (26.5 ± 10.9)-IQR 3.7			
Door-to-needle time (DNT) in minutes	<mean (61.7)	222	225	447
	≥mean (61.7)	159	117	276
	Mean ± SD (61.7 ± 40.1)-IQR 40			
Hospital-Acquired Pneumonia	0: No	375	292	667
	1: Yes	6	50	56
UTI	0: No	378	310	688
	1: Yes	3	32	35
Ischemic stroke subtype (TOAST)	Large Vessel Disease (LVD)	101	137	238
	Cardioembolism (embolic)	102	76	178
	Small Vessel Disease (SVD)	104	73	177
	Stroke of other Determined Etiology (SDO)	53	24	77
	Stroke of Undetermined Etiology (SUO)	21	32	53
90-day mRS		381	342	723

**Table 2 jpm-13-01555-t002:** Machine learning model performance.

Model	Accuracy	Precision	Specificity	Recall	F1-Score	AUC	MCC	Log Loss	Brier Score
XGB Classifier	0.676	0.569	0.648	0.719	0.636	0.756	0.359	0.761	0.235
Random Forest Classifier (RF)	0.683	0.580	0.670	0.702	0.635	0.758	0.364	0.584	0.200
Support Vector Machine (SVM)	0.697	0.600	0.705	0.684	0.639	0.762	0.382	0.594	0.201
Logistic regression (LR)	0.655	0.549	0.636	0.684	0.609	0.719	0.313	0.636	0.220
Decision Tree Classifier (CART)	0.621	0.514	0.614	0.632	0.567	0.623	0.240	13.672	0.379

## Data Availability

The datasets generated and/or analyzed during the current study are available from the corresponding author on reasonable request and subject to appropriate ethical approvals.

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
