# Peer review of "Predicting 90-Day Prognosis in Ischemic Stroke Patients Post Thrombolysis Using Machine Learning"

_jpm, 2023, doi:10.3390/jpm13111555_

Round 1

Reviewer 1 Report

Generating a machine learning model on the basis of a retrospective study with missing data is the biggest pitfall of the study . Translating and interpreting  this in a clinical context  is difficult . Although the machine learning  models generated are accurate statistically & seem attractive; yet  they seem to add little to the current  clinical armamentarium. All the modifiable  parameters are routinely treated stroke care specialists in specific clinical settings.

Author Response

Response: We appreciate your thoughtful feedback on our study. The percentage of missing data is less than 6% which, compared to many other studies in the literature (8%-24%), is considered low [1, 2]. To account for the missing data, we have utilized multiple imputations using the chained equations (MICE), considered one the most up-to-date and trusted methods to handle missing data [3]. (L204-210)

It is important to note that missing data is inevitable when training and developing machine learning models using real-world data. Your concerns are valid, and we acknowledged their challenges in our limitations and tried to counter them with the best approach possible. We agree that translating machine learning results into a clinical context can be challenging. Our intention in this study was to demonstrate the feasibility and potential of using machine learning in stroke care, with the hope that it could inspire further research in this area, as we stated in our study. We recognize the importance of validating our findings in prospective clinical settings, which is a logical next step. With regards to the value added to the current clinical armamentarium, we understand your concern about the perceived limited additional value of the machine learning models compared to current clinical practices. However, we believe that machine learning models have the potential to enhance clinical decision-making in several ways, such as identifying subtle patterns or predicting outcomes that may not be apparent with traditional methods. Further research and validation in real-world clinical settings are necessary to fully assess their utility, as we pointed out in the limitations and the conclusion. Finally, regarding modifiable parameters, our aim is not to duplicate existing efforts but to identify areas where machine learning can complement and enhance the current clinical practices.

References:

  1. Lolak S, Attia J, McKay GJ, Thakkinstian A (2023) Comparing Explainable Machine Learning Approaches With Traditional Statistical Methods for Evaluating Stroke Risk Models: Retrospective Cohort Study. JMIR Cardio 7:e47736. https://doi.org/10.2196/47736
  2. Li R, Harshfield EL, Bell S, et al (2023) Predicting incident dementia in cerebral small vessel disease: comparison of machine learning and traditional statistical models. Cereb Circ - Cogn Behav 5:100179. https://doi.org/10.1016/j.cccb.2023.100179
  3. Azur MJ, Stuart EA, Frangakis C, Leaf PJ (2011) Multiple imputation by chained equations: what is it and how does it work? Int J Methods Psychiatr Res 20:40–49. https://doi.org/10.1002/mpr.329

Reviewer 2 Report

Predicting 90-Day Prognosis in Ischemic Stroke Patients Post-Thrombolysis Using 2 Machine Learning

A very interesting work with a very important number of stroke patients undergoing thrombolytic therapy, had the objective of constructing a machine learning model for predicting the prognosis 40 of ischemic stroke patients who underwent thrombolysis, assessed through the modified Rankin Scale (mRS). score 90-day after discharge. From a Registry from Qatar’s stroke registry covering January 2014 and June 2022. A total of 723 patients with ischemic stroke who had received thrombolysis were included. Support Vector Machine model emerged as the standout performer, achieving an area under the curve of 0.72. Key determinants of patient outcomes included stroke severity at admission Emphasizes the importance of utilizing machine learning for early prognosis prediction in ischemic stroke management, particularly post-thrombolysis care. -I think the introduction is a long (some of the concepts raised in the introduction could lead to conclusions), the objectives are very well stated, the results and conclusions are obtained that are consistent with them. Predicting the prognosis of ischemic stroke. prioritizes the importance of early assessment, leveraging machine learning models such as SVM and adapting care plans to individual patient needs. -I think that the results obtained in figures 3a and 3b should appear in the “Results” section. Very interesting, the study contributes to our knowledge and encourages future research to incorporate a more complete data integration. Ultimately, it offers a path forward in the quest to improve personalized stroke care and ultimately improve the quality of life for stroke survivors.

Author Response

Dear reviewer, many thanks for your insightful comments. 

Reviewer 3 Report

Dear Authors,

Kindly address the following.

1. Can you justify the need of inclusion of patients from different countries?

2. In regard to Mode of arrival,- Any particular reason to mention this aspect. Whether this aspect effected any results?

3. Any other pre-existing co-morbities has been observed in the patients, other than mentioned in the article?

4. The inclusion and exclusion criteria of number of patients is not clear

5. XGBoost classifier, Random Forest (RF), Support Vector Machine (SVM), Logistic Regression (LR), and Classification and Regression Tree (CART).- Why only these five machine learning models were trained? Need to mention clearly about this aspect.

6. Discussion needs to be explain more in terms of patient management and other affected factors.

Author Response

Dear Authors,

Kindly address the following.

1. Can you justify the need of inclusion of patients from different countries?

Response: Extensive literature underscores the significant role ethnicity plays in determining disease prognosis and treatment outcomes. It is well-established that genetic, environmental, and socio-cultural factors associated with different ethnic groups can influence the course of various medical conditions. Understanding these variations is paramount for tailoring healthcare strategies effectively, especially in a multi-ethnic healthcare context like Qatar. Also, the inclusion of patients from different countries in our study is justified because Qatar's diverse population, with Qataris as a minority, necessitates gaining a deeper understanding of the healthcare landscape and patient needs in Qatar, enhancing the relevance of our findings to the local healthcare context. It turns that the ethnicity in this study has little impact on the overall performance of the model, but this has been reached aftermath.

2. In regard to Mode of arrival,- Any particular reason to mention this aspect. Whether this aspect effected any results?

Response: In managing stroke, it is established that the speed of access to specialized stroke care is paramount for saving the brain cells minimizing the risk of unfavorable outcomes. It is reasonable to consider that patients who arrive via ambulance may receive skilled care during the journey and may arrive faster to the hospital which improve their chances compared to those that arrive to the hospital via private vehicles. Also, this variable had a minimal impact on the model’s performance. Therefore, we haven’t elaborated in explaining its impact on the outcome in the discussion.

3. Any other pre-existing co-morbities has been observed in the patients, other than mentioned in the article?

Response: The dataset which we have obtained from the stroke registry includes the comorbidities that we disclosed in the study as in table 1. In addition, it includes other comorbid conditions such as renal failure, congestive heart failure and epilepsy that had very few patients diagnosed with the condition (in some zero patients) after we carried out the exclusion criteria as illustrated in figure 1. This makes the variable of no value and skews the data significantly. Therefore, where we had very few cases diagnosed with the condition, the variable was excluded.  

4. The inclusion and exclusion criteria of number of patients is not clear

Response: As illustrated in Figure 1, we started with all patients in the dataset. We target patients who were diagnosed with ischemic stroke and went through thrombolysis which resulted in 971 patients. Since our outcome variable is 90-day mRS post discharge, patients who haven’t made it to the follow-up appointment due to death or other unknown reason were excluded resulting in 723 eligible cases.

5. XGBoost classifier, Random Forest (RF), Support Vector Machine (SVM), Logistic Regression (LR), and Classification and Regression Tree (CART). -Why only these five machine learning models were trained? Need to mention clearly about this aspect.

Response: Thank you for your comment, we appreciate your inquiry and would like to provide a clear explanation for why we chose these specific models and address your concern. In our paper, the rationale for choosing these specific models was that they are widely recognized for their computational efficiency and frequently used classifiers across various domains, especially medical and classification studies. Also because of their relevance to the topic, diversity, interpretability, and benchmarking capabilities this opens the door for future comparability. In addition, we experimented with other classifiers, but these were the best performing. Additionally, we will acknowledge that while these models offer a solid starting point, future research may explore additional algorithms or ensemble techniques to further improve predictive performance.

  1. Discussion needs to be explain more in terms of patient management and other affected factors.

Response: The aim of the study is to identify the factors that enhance the predictability of the 90-day prognosis. Interestingly, all the powerful variables turned to be collected during the initial visit of the patient which enhances the predictability of the prognosis. The scope of the study is not to enter the stroke clinical management per se which is standard to a great degree. Instead, we aim to inform the clinicians that having x, y, and z factors increases the chance of having unfavorable outcomes, so they have the liberty to work within the evidence-based boundaries to set adequate preventive measures. Until this stage, machine learning hasn’t reached to a stage to prescribe the treatment, but it may help provide some insights to the clinicians to modify and personalize the care plans for the best interest of the patients.

Reviewer 4 Report

This paper “Predicting 90-Day Prognosis in Ischemic Stroke Patients” aimed to construct a machine learning model for predicting the prognosis of ischemic stroke patients who underwent thrombolysis, assessed through the modified Rankin Scale (mRS) score 90-day after discharge. Data were sourced from Qatar’s stroke registry covering January 2014 and June 2022. A total of 723 patients with ischemic stroke who had received thrombolysis were included. Clinical variables were examined, encompassing demographics, stroke severity indices, comorbidities, laboratory results, admission vital signs, and hospital-acquired complications. The predictive capabilities of five distinct machine learning models were rigorously evaluated using a comprehensive set of metrics. The SHAP analysis was deployed to uncover the most influential predictors. Support Vector Machine model emerged as the standout performer, achieving an area under the curve of 0.72. Key determinants of patient outcomes included stroke severity at admission, admission systolic and diastolic blood pressure, baseline comorbidities, notably hypertension and coronary artery disease, stroke subtype, particularly strokes of undetermined origin, and hospital-acquired urinary tract infections.

The topic is justified. The paper could be further improved if the following remarks are taken into consideration:

1.       ABSTRACT: overall abstract should be concise, especially the conclusion portion of the abstract.

2.       A few of the grammatical mistakes were found in the whole draft of the article.

3.       Better to name the section ‘Background’ as an ‘Introduction’.

4.       Currently ‘Background’ section lacks justification of the research. The contribution may be key fold in the ‘Background’ section.

5.       Although, the background section is well supported with the facts and figures, however, Qatar is not so populated, strangely, 15,859 patients sought specialized stroke care, and 723 patients were ultimately included in the study.

6.       Explicitly need to write in number that how many ‘Baseline variables’ are used in the study?

7.       Explicitly need to state in number about training and validation split about each of the five groups based on their stated nationality, i.e., Qatari, Middle East and North Africa (MENA) region, South Asia region, Southeast Asia region (following the United Nations geo-scheme), and all other nationalities grouped as 'other' (27-29).

8.       Dataset: as I can guess data is imbalanced for training or validation purpose for each of the machine learning models used for model training.

9.       Most of the ML models, i.e., XGBoost classifier, RF, SVM, LR, and CART yield accuracy near to the baseline accuracy, which is not acceptable technically.

10.   What these evaluation measures, i.e., MCC, log loss, and Brier score are representing in connection with ‘Ischemic Stroke’ prognosis?

11.   The motivation is not clear. Please specify the importance of the proposed solution. Is that able to be deploy in real-time environment at Hamad Medical Corporation, Doha, Qatar? Will medical practitioners (even at Hamad Medical Corporation, Doha, Qatar) accept this module with 69.7% accuracy?

need minor improvements

Author Response

This paper “Predicting 90-Day Prognosis in Ischemic Stroke Patients” aimed to construct a machine learning model for predicting the prognosis of ischemic stroke patients who underwent thrombolysis, assessed through the modified Rankin Scale (mRS) score 90-day after discharge. Data were sourced from Qatar’s stroke registry covering January 2014 and June 2022. A total of 723 patients with ischemic stroke who had received thrombolysis were included. Clinical variables were examined, encompassing demographics, stroke severity indices, comorbidities, laboratory results, admission vital signs, and hospital-acquired complications. The predictive capabilities of five distinct machine learning models were rigorously evaluated using a comprehensive set of metrics. The SHAP analysis was deployed to uncover the most influential predictors. Support Vector Machine model emerged as the standout performer, achieving an area under the curve of 0.72. Key determinants of patient outcomes included stroke severity at admission, admission systolic and diastolic blood pressure, baseline comorbidities, notably hypertension and coronary artery disease, stroke subtype, particularly strokes of undetermined origin, and hospital-acquired urinary tract infections.

The topic is justified. The paper could be further improved if the following remarks are taken into consideration:

  1.      ABSTRACT: overall abstract should be concise, especially the conclusion portion of the abstract.

Response: the conclusion part has been revised as suggested. L52-57

  1.      A few of the grammatical mistakes were found in the whole draft of the article.

Response: the entire manuscript has been revisited.

  1.      Better to name the section ‘Background’ as an ‘Introduction’.

Response: revised as suggested. L70

  1.      Currently ‘Background’ section lacks justification of the research. The contribution may be key fold in the ‘Background’ section.

Response: revised as suggested. L148-155

  1.      Although, the background section is well supported with the facts and figures, however, Qatar is not so populated, strangely, 15,859 patients sought specialized stroke care, and 723 patients were ultimately included in the study.

Response: Thanks for the comment. That is true! Qatar’s population is little above 2.5 mil. This high number of people who sought care in the stroke center is a result of two factors.

-           There is only one tertiary hospital that provides the specialized stroke care in the country. Therefore, this number (15859) is the total of more than 9 years that are covered in the study.

-           Secondly, the database includes ischemic stroke, hemorrhagic stroke in addition to the TIA and mimics that make a significant number of the 15859 cases.

Regarding the final number of eligible patients, this is the result of the inclusion criteria that focuses on patients who underwent thrombolysis therapy (971) and had their 90-day follow-up appointment so their 90-day mRS is captured in the database which is 723 patients (explained in figure 1). 

  1.      Explicitly need to write in number that how many ‘Baseline variables’ are used in the study?

Response: thank you, we have utilized 28 variables in the model development. The parameters are outlined in Table 1. L197-198

  1.      Explicitly need to state in number about training and validation split about each of the five groups based on their stated nationality, i.e., Qatari, Middle East and North Africa (MENA) region, South Asia region, Southeast Asia region (following the United Nations geo-scheme), and all other nationalities grouped as 'other' (27-29).

Response: Thank you for your comment, please find below the number of patients from each nationality for both training and test data. However, since all variables were split into training and test sets, we don’t see any value added if we present that level of granular detail only for this variable.

Train Data

S.Asia

302

MENA

119

Q

74

S.E Asia

54

Other

29

Test Data

S.Asia

78

MENA

32

Q

16

S.E Asia

14

Other

5

  1.      Dataset: as I can guess data is imbalanced for training or validation purpose for each of the machine learning models used for model training.

Response: thank you for your valuable comment, we have tackled this problem at multiple levels. At the preprocessing level, we used stratified random sampling, to ensure that the class ratio is maintained during training. At the training level, we included ensemble methods such as XGboost classifier, furthermore we have utilized class weighting to ensure that the class frequency does not skew the results. Finally, for model evaluation, we used suitable evaluation metrics that are shown to be good in case of data imbalance such as MCC and f1 score, which give a complementary view aside with other metrics.

  1.      Most of the ML models, i.e., XGBoost classifier, RF, SVM, LR, and CART yield accuracy near to the baseline accuracy, which is not acceptable technically.

Response: thank you for raising this valuable point, while achieving high accuracy is desirable, the clinical relevance of a model extends beyond accuracy alone. Other evaluation metrics are also crucial factors. The result of our models showed a stable performance across reported metrics. Our results show a promising ground for future improvement.

  1.  What these evaluation measures, i.e., MCC, log loss, and Brier score are representing in connection with ‘Ischemic Stroke’ prognosis?

Response: Thank you for your question, MCC (Matthews Correlation Coefficient) is a measure of the quality of binary classifications. It takes into account true positives (TP), true negatives (TN), false positives (FP), and false negatives (FN) to provide a balanced assessment of classification performance. It complements the f1 score. MCC is useful in assessing how well a classification model, such as a prognosis model for ischemic stroke, balances between correctly identifying cases and non-cases while considering both false positives and false negatives. A higher MCC indicates a more accurate and balanced model. Log loss measures the performance of a classification model by quantifying the uncertainty of its predictions. It penalizes predictions that are confidently wrong and encourages well-calibrated probability estimates. Log loss is relevant for prognosis models because it provides a measure of how well the model's predicted probabilities align with the actual outcomes. For 'Ischemic Stroke' prognosis, lower log loss implies more confident and accurate predictions of stroke risk.

Finally, the Brier score assesses how well predicted probabilities match the actual outcomes. it is relevant in the context of prognosis for 'Ischemic Stroke' because it evaluates the calibration and accuracy of predicted probabilities. Lower Brier scores indicate that the model's predicted probabilities closely match the observed outcomes, which is essential for an accurate prognosis.

  1.  The motivation is not clear. Please specify the importance of the proposed solution. Is that able to be deploy in real-time environment at Hamad Medical Corporation, Doha, Qatar? Will medical practitioners (even at Hamad Medical Corporation, Doha, Qatar) accept this module with 69.7% accuracy?

Response: We greatly appreciate your valuable feedback. To address your comment on motivation, we've incorporated this aspect into the background section (L148-155) to provide a comprehensive context that underscores the significance of our study.

Regarding the model's accuracy, it's imperative to recognize that the application of machine learning in healthcare is still in its early stages despite the plethora of published studies, and our model's performance is not static but rather dynamic. Over time, we envision improving its accuracy through continual training, rigorous contextual validation, and the incorporation of more pertinent and impactful variables, as we have delineated in our limitations and conclusion sections.

In our study, we have openly acknowledged our limitations, which encompass the imperative need for enhanced variable inclusivity and the ongoing imperative to fortify our database and registry. This aspect represents a substantial contribution to our research, not only within our local setting at Hamad Medical Corporation, Qatar, but also within the broader global healthcare domain. As machine learning models undergo continuous refinement and advancement, their acceptance by medical practitioners, even at Hamad Medical Corporation, hinges on their consistent ability to exhibit heightened accuracy, reliability, and practical utility in real-world scenarios. To this end, we are fully committed to progressing towards this objective. We aim to achieve this by transparently emphasizing the importance of validating our model within real clinical settings and by accentuating the necessity of including critical variables within the stroke database to ensure the utmost effectiveness of our approach.